# Autoreactive T-Cells in Psoriasis: Are They Spoiled Tregs and Can Therapies Restore Their Functions?

**DOI:** 10.3390/ijms24054348

**Published:** 2023-02-22

**Authors:** Immacolata Pietraforte, Loredana Frasca

**Affiliations:** 1Department of Oncology and Molecular Medicine, Istituto Superiore di Sanità, 00161 Rome, Italy; 2National Center for Drug Research and Evaluation, Istituto Superiore di Sanità, Viale Regina Elena, 299, 00161 Rome, Italy

**Keywords:** autoreactivity, Tregs, psoriasis

## Abstract

Psoriasis is a chronic inflammatory skin disease, which affects 2–4% of the population worldwide. T-cell derived factors such as Th17 and Th1 cytokines or cytokines such as IL-23, which favors Th17-expansion/differentiation, dominate in the disease. Therapies targeting these factors have been developed over the years. An autoimmune component is present, as autoreactive T-cells specific for keratins, the antimicrobial peptide LL37 and ADAMTSL5 have been described. Both autoreactive CD4 and CD8 T-cells exist, produce pathogenic cytokines, and correlate with disease activity. Along with the assumption that psoriasis is a T-cell-driven disease, Tregs have been studied extensively over the years, both in the skin and in circulation. This narrative review resumes the main findings about Tregs in psoriasis. We discuss how Tregs increase in psoriasis but are impaired in their regulatory/suppressive function. We debate the possibility that Tregs convert into T-effector cells under inflammatory conditions; for instance, they may turn into Th17-cells. We put particular emphasis on therapies that seem to counteract this conversion. We have enriched this review with an experimental section analyzing T-cells specific for the autoantigen LL37 in a healthy subject, suggesting that a shared specificity may exist between Tregs and autoreactive responder T-cells. This suggests that successful psoriasis treatments may, among other effects, restore Tregs numbers and functions.

## 1. Introduction

Psoriasis is a chronic inflammatory skin disease affecting 2–4% of the global population [1]. Psoriasis has a strong genetic component, but environmental factors are also important [2,3]. Psoriasis is characteristic, with thickening and scaling of the epidermis due to hyper-proliferation of keratinocytes (acanthosis). CD4 and CD8 T-cell infiltrate characterizes the skin in psoriasis, a T-cell-driven disease, but other cell types are also present in the lesions, including neutrophils, macrophages, NK cells, dendritic cells [4]. It has been long known that the pathogenesis of psoriasis is driven by T-cell derived factors, produced by subsets including T-helper (Th) 1 cells, Th17, Th22 and regulatory T cells (Tregs). There is evidence in animal models that Tregs could play a role in ameliorating psoriasis [5]. However, these Tregs are considered dysfunctional, as they are altered in frequency, phenotype, and function [6]. It is possible that Th17-cells could be diverted Tregs in psoriasis. If this is true, pharmacological intervention could restore their functions. This is an important goal to achieve in therapy, to stabilize the disease after treatment, which could also give the opportunity to discontinue therapy without unwanted consequences. Tregs are of various types: they can act through secretion of regulatory/suppressive cytokines like IL-10, TGF-β, IL-35 or act by contact. Tregs that express Foxp3 can lose their capacity to suppress in certain circumstances. For instance, Tregs can convert to IL-17-like cells under inflammatory conditions, especially in the presence of IL-6, which is a cytokine highly up-regulated in psoriasis [7]. In the last part of this narrative review, we report that a healthy person can harbor the T-cell specific for LL37, which are able to proliferate and make cytokines in response to LL37 stimulation. We show the existence of a Tregs counter-part of the LL37-specific T-cell pool, endowed with suppressive functions in vitro.

## 2. Tregs Overview

### 2.1. Tregs Subsets in Health and Disease

Work with animal models has shown that more Tregs subsets exist, which parallel the T-helper cell subsets [8]. Most of these studies were in mice, as analogous subsets in humans are more difficult to study because of lack of accurate identification markers [9,10].

Tregs expressing the Th1 factor T-bet exist, which also express chemokine receptor CXCR3 [11]. The deletion of Foxp3 in T-bet-expressing cells was shown to determine uncontrolled Th1 immune responses at steady state [12]. In addition, Th2-like Tregs have been identified, for instance, in non-lymphoid organs. The Th2-like Tregs can express GATA-3 and have been detected at mucosal surfaces: skin, gut, and lungs [13,14]. Indeed, studies demonstrated that, in mice, GATA-3 expression is required for Tregs stability and maintenance of Foxp3 expression [15]. A pathogenic reprogramming of T-cells can cause disease, and there are several examples of this. For instance, Th2 Tregs can play a role in food allergies [16]. Relevant to psoriasis, Tregs can also express the canonical Th17 transcription factor, RORγt. Cells of these types have been detected in the gut [17,18], but they have been also described in patients with psoriasis [19], as well as in inflammatory bowel diseases (IBD), and arthritis; in some cases, they convert into IL-17 secreting cells [20,21,22,23].

Single cell RNA sequence analysis shows the existence of distinct T-cell subsets in relapsing psoriasis for tissue-resident T-cells. Th17 cells and NF-kB signaling pathways evidenced include Pellino-1 (PELI1). Mice with systemic and conditional depletion of PELI1 were protected from psoriasiform dermatitis and showed reduced IL-17A production and NFkB activation in Th17 cells. The inhibition of PELI1 significantly ameliorated murine psoriasiform dermatitis by reducing IL-17A production. PELI1 belongs to a member of the E3 ubiquitin ligases, mediating immune receptor signaling cascades involving, among others, the NFkB pathway; it also promotes intrinsic activation of skin resident Th17 cells in psoriasis. This points out that this gene inhibition could be a promising therapeutic strategy for psoriasis, limiting skin-resident Th17 cell responses. Both the T-cell receptor (TCR) and IL-23 stimulation upregulate PELI1 expression in Th17 cells from lesional skin. However, it is currently unknown whether PELI1 directly modulates the suppressive function of Tregs [24].

Recently, the regulatory counterpart of follicular helper T-cells (Tfh) cells, which promote germinal center (GC) responses and antibody production, has been described. These cells express TCF1 and LEF1, required for expression of B-cell lymphoma 6 (Bcl-6) [8].

### 2.2. Tregs in the Skin and in Psoriasis

It has been shown that Tregs are part of the TRM (Tissue Resident Memory cells) pool, and it is postulated that these cells can re-circulate. Human psoriatic plaques contain abundant numbers of IL-17A-producing T-cells with a CD69+ and/or CD103+ TRM phenotype. TRM can play a role in chronic inflammatory diseases of the skin, such as vitiligo and psoriasis, and are likely involved in recurrent lesions. A greater number of CD8 TRM cells infiltrate psoriatic lesional and non-lesional skin, as compared to normal skin. These are mainly infiltrating CD8+IL-17A+ TRM cells and correlate with disease duration. Apparently, healed skin contains CD8 TRM cells, which mostly localize in the epidermis (as they express CD103), while CD4 TRM cells localize in the dermis. Moreover, a large pool of TRM cells express IL-17A alone, IL-17A and IL-22, or IL-22 alone, and persist in post-lesional psoriasis skin. Cytokine production is not exclusively oriented toward a Th17 phenotype as TRM also express IFN-γ, showing a Th1 phenotype, which could be induced by IL-15 stimulation [25].

Tregs residing in the skin seem primary localized, in mouse and humans, in the hair follicles (HFs) [22,26,27]; this is due to HF epithelial cells production of CCL20, which recruits CCR6-expressing Tregs into the skin [26]. In human and mice skin, Tregs constitute about 20–40% of the CD4 T-cells [22]. These Tregs can express GATA-3 [15]. Their role is to maintain the cutaneous immune homeostasis, promote wound healing, and repair tissues. Interestingly, they express ST2 [28], the receptor for IL-33, which is released in situation of tissue damage [29]. IL-33 is an alarmin, which can regulate Foxp3 in the mucosal tissue [30]. IL-33 was originally described as a cytokine that activates Th2 cells, but it can affect various cell types, including Th1 cells or innate lymphoid cells (ILC) and CD8 T-cells [31]. Its exact role in psoriasis is not completely clear due to these pleiotropic effects.

In support of the idea that Tregs increased because of tissue damage stimuli, a recent study showed that Tregs in the skin expand following UVB irradiation [32]. After injury, Tregs express the AREG receptor, EGFR, suggesting an AREG autocrine role in tissue repair [33]. The peripheral blood of healthy humans contains Tregs, which express cutaneous lymphocyte antigen (CLA) and other skin homing receptors [34], suggesting that Tregs migrate to the skin using these unique receptors. Injury and microbial invasion influence this migration. Similar to human Tregs, mouse Tregs express some homing receptors, which facilitate their recruitment into inflamed skin [35].

Mouse and human studies indicate that, in psoriasis, the IL-23/IL-17 axis of inflammation, together with Tregs dysfunction, determines the Th17/Tregs imbalance implicated in the disease [6,36], although the link between Tregs and disease severity is debated [37]. The conflicting results may depend on several factors: the site of biopsies, psoriasis subtypes, and different disease status. Although Tregs and Th17 cells increase in adult and pediatric psoriasis patients, Tregs were shown to be unable to suppress Th17 cells activation, and their effector T-cell responses and proliferation [6,37].

Several papers support Tregs dysfunction in psoriasis. The Tregs suppressive function seem impaired due to the pro-inflammatory cytokine milieu. For example, the exposure to high level of IL-6 decreases Tregs activity [7,38]. Dysfunctional Tregs in peripheral blood of patients with psoriasis have been reported showing that they have a phosphorylation and an aberrant activation of STAT3, which is due to the effects of pro-inflammatory cytokines, not only IL-6, but also IL-21 and IL-23 [39]. Additionally, inhibition of Foxp3 by up regulation of microRNA (mir-210), in CD4 T-cells, results in decreased levels of IL-10 and TGF-β by these cells, accompanied by increased levels of pro-inflammatory cytokines release, such as TNF-α and IL-17A, as described by Zhao et al. [40].

Tregs can express CD39 and CD73, so they can use the adenosine signaling for exerting suppressive effects. Yan et al. reported that Tregs from patients with psoriasis have a reduced expression of CD73 and a CD73/AMPK pathway deactivation [41]. The adenosine pathway (CD39 and CD73) is widely known to play a crucial role for the Tregs immunosuppressive function. CD39 removes extracellular ATP by hydrolyzing ATP/UTP and ADP/UDP into AMP. Successively, AMP is rapidly degraded into adenosine in the presence of CD73. Then, adenosine, which binds to the A2AR receptor on Tregs, triggers the accumulation of intracellular cAMP, activation of AMPK and inactivation of mTOR, blocking IL-17 and IFN-γ production. This also blocks the differentiation of Tregs into IL-17 secreting Tregs, which have a decreased immune suppressive function. Moreover, adenosine binding to A2AR of the T effector cells, achieves local repression of immune response and down regulation of INF-γ production.

Inefficient recruitment of Tregs to inflamed skin can also concur to the inefficiency in restraining inflammation in individuals with psoriasis [42]. Finally, although microbial infection in the skin is a contributing factor in psoriasis, a connection by skin microbiota and Tregs in psoriasis can be hypothesized [43]. All these data in the scientific literature show that the suppressive function of circulating and in skin-resident Tregs is important in psoriasis and support an impairment of Tregs in the psoriatic disease.

### 2.3. Tregs Plasticity in Psoriasis

Tregs and Th17 cells require TGF-β to develop from a common precursor, indicating a constant competition between the two cell types [44], highlighting a special relationship in the development and function of Th17 and Tregs. Several studies revealed that a subset of Tregs in the skin might differentiate into Th17 cells [19,45]. It seems that CD27 and OX40 expression on Tregs plays a role in suppressing Tregs differentiation toward Th17 phenotype, whereas lack of expression of these molecules induces the expression of high levels of IL-17A and the transcription factor RORα [46,47]. Tregs from patients with psoriasis are able to differentiate into IL-17A producing cells after stimulation ex vivo. Tregs may also differentiate into IL-17A producing cells upon activation of the histone deacetylase 1 (HDAC-1), which is elevated in psoriatic lesions. Moreover, the presence of IL-17A^+^Foxp3^+^CD4 T-cells was observed in psoriatic skin [19]. All these data show not only a Tregs dysfunction but also a phenotypic alteration of these cells in psoriasis.

### 2.4. Treatments That Can Affect Tregs in Psoriasis

Many treatments are in use for psoriasis, and others are being explored. An overview on how these treatments appear to influence expansion or functionality of Tregs and the Th17/Tregs balance is reported below. The therapeutic approach to psoriasis comprises two major categories of drugs: biological agents, and immunosuppressive drugs (methotrexate, cyclosporine), and treatments sub-groups are evidenced.

#### 2.4.1. Biological Agents

##### TNF-α Antagonists

Among biological agents, TNF-α antagonists have been extensively used. These include infliximab (a chimeric monoclonal antibody composed of a human IgG1 constant region and a murine variable region), etanercept (a soluble TNFR, made of two extracellular domains of the human TNFR2 fused to the Fc fragment of human IgG1), and adalimumab (a humanized monoclonal antibody). Different studies in humans refer that anti TNF-α agents increase Tregs and decrease the Th17 cells frequency in peripheral blood of psoriasis patients [48,49,50]. Etanercept regimen showed a more significant modification of the T-cell subsets, as compared to the other two drugs [51,52,53]. Moreover, Diluvio et al. reported that infliximab treatment induces a polyclonal expansion of Tregs, sorted from peripheral blood of patients, showing a diverse TCR repertoire [54]. In all of these studies Tregs suppressive function was not addressed and the modification of the subset of Tregs in the skin has not been analyzed; the data refer only to peripheral blood cells. Of note, in a murine psoriasiform model, some data are conflicting, and not confirmed in humans [55].

##### IL-17 and IL-23 Antagonists

Classes of biologic agents targeting either IL-17 or IL-23 demonstrated higher rates of response and superiority compared to previous biologic agents. Among these new therapeutic agents are an anti-IL-17A (secukinumab), an anti-IL-23p19, called guselkumab, and an anti-p40, called ustekinumab. In a model of imiquimod-induced psoriasis in mice, it was described an increase of Foxp3^+^ Tregs in the skin and restoration of their suppressive function following use of IL-17 or IL-23 blocking antibodies but not with an anti-TNF-α treatment [55]. Of note, Kanman et al. described a principal role of IL-23 in regulation of Tregs plasticity and conversion into a Th17 like phenotype [56]. These data are supported by observations in humans that IL-23 inhibitors act as potent disease modifying drug, more than IL-17 antagonists [57,58,59]. Clinical trials testing IL-23 inhibitors showed long-lasting maintenance of the therapeutic response following treatment discontinuation, compared to IL-17 inhibitors [58,60,61]. Another clinical trial, comparing guselkumab to secukinumab, provided relevant insights about the skin compartment. During a 24-week treatment with guselkumab or secukinumab, the number of CD4 and CD8 TRM cells decreased in psoriatic lesions of both treatment arms, but guselkumab reduced memory T-cells, maintaining Tregs whereas the opposite was observed for secukinumab treatment [57]. Secukinumab treatment decreased the number of Tregs in a more pronounced way than guselkumab. Moreover, a greater decrease of LC (Langerhans cells), infiltrating post-lesional skin, was observed after IL-23 blockade. These findings suggest a successful response to either IL-23 or IL-17 inhibitors, with an increased Tregs/CD8 TRM ratio. A superior long-term control of skin inflammation was achieved by inhibiting IL-23 with a reset of the pathogenic inflammatory T-cells and an increase of Tregs. IL-23 acts on the T-cell compartment and stimulates the expression of RORγt, and the production of IL-17A, IL-1F, and IL-22. Moreover, IL-23 drives and maintains the differentiation of Th17 cells. In contrast, IL-17A is an effectors cytokine that induces skin inflammation. It is also expressed by neutrophil and mast cells and produces indirect effect on the T-cell compartment. This may explain why IL-17 inhibitors have lower modulator ability than IL-23 inhibitors.

##### IL-6 Antagonists

IL-6 is another important cytokine involved in psoriasis. Although IL-6 plays a role in the maturation of Th17 cells, search of the literature and clinical trials in websites did not reveal psoriasis studies with anti-IL-6/IL-6 receptors and role on Tregs frequency and Th1/Tregs balance [62]. A study in vitro reported that IL-6 was necessary and sufficient to reverse human T-cell suppression by Tregs in in vitro models using activated DCs as a source of IL-6 [7]. Although IL-6 may be another potential target for psoriasis treatment, data in the literature show that attempts to treat psoriasis with tocilizumab (TCZ), a humanized anti-interleukin-6 (IL-6) receptor antibody licensed for the treatment of rheumatoid arthritis (RA), have been unsuccessful [63]. On the other hand, the use of new IL-6 inhibitors such as clazakizumab, a monoclonal antibody with high affinity and specificity for IL-6, could be more promising for psoriatic arthritis (PsA) [57,64,65,66].

#### 2.4.2. Photo Therapy

Many photo therapeutic approaches can treat psoriasis: natural phototherapy, broadband UVB, narrowband UVB, selective UV phototherapy, Xenon chloride excimer laser, Xenon chloride excimer lamp, UVB light emitting diode, flat-type fluorescent UVB lamp, UVA, Mixed UVB/UVA, Psoralen+ UVA photochemotherapy (PUVA), bath water delivery of 8-methoxypsoralen and subsequent UVA-irradiation (bath-PUVA therapy), UVA-1 phototherapy, Pulsed Dye laser, and others [67,68]. The principal immunomodulatory effect of phototherapy is promoting the death of effector cells, such as T-cells, and keratocytes, and inhibition of LC, macrophages, neutrophils and NK cell function. Effect of bath-PUVA therapy was reported on three distinct Foxp3^+^ subsets: activated Tregs (aTregs), resting Tregs (rTregs), and cytokine-secreting non-suppressive T-cells from peripheral blood of psoriasis patients and healthy controls. Bath-PUVA therapy increased Tregs and restored dysfunctional Tregs activity in patients: in particular aTregs were significantly increased in the early bath-PUVA therapy sessions, and then diminished. RTregs, which were lower in patients than healthy controls, increased during therapy [68]. Takuya Furuhashi et al. confirmed these data by functional assays. CD4 CD25^−^ cells separated from PBMCs of psoriasis patients treated with PUVA and activated with anti-CD3/CD28-bound beads, were cultured with or without CD4 CD25^+^ T-cells. The ability of Tregs to suppress CD4 CD25^−^ T-cells was calculated by comparing the proliferation rates of CD4 CD25^−^ T-cells in the presence/absence of CD4 CD25^+^ T-cells [69]. The same conclusions derive from an UV (B) treatment in psoriasis patients with polymorphic light eruption (PLE), in which UV increased the number of Tregs. This might be a compensatory mechanism to counteract the susceptibility to PLE [70]. Tregs from patients with PLE lacked any capacity to suppress effector T-cell proliferation but this capacity improved after therapy, as demonstrated by regulatory T-cell suppression assay. Moreover, after UVB treatment, keratinocytes upregulated the expression of receptor of activated nuclear factor-B ligand (RANKL). This receptor interacts with RANK on DCs, making DCs able to expand the number of Tregs [71,72]. These data were confirmed also in a mouse model of psoriasis [73].

#### 2.4.3. Vitamins

##### Vitamin A

Vitamin A derivatives, retinoids, are also of common use to treat psoriasis. Retinoids, such as etretinate or acitretin, are absorbed in the small intestine and then are metabolized in other organs to the active acid form of retinoid acids (RAs), which interact with retinoid X receptors (RXRs). This heterodimer binds the RA response element on CNS1 of Foxp3, inducing Foxp3 expression and the generation of peripheral Tregs from naive T-cells [74]. Of note, retinoids not only promote Tregs generation but also regulate TGF-β, capable of inhibiting the IL-6. IL-6 is driving activation of pro-inflammatory Th17 cells, acting on RORγt [75,76].

##### Vitamin D

Vitamin D seems to regulate Tregs. Vitamin D status correlates with circulating Tregs in patients affected by psoriasis; a correlation with the severity of the disease, evaluated with Psoriasis Area Severity Index (PASI) score is present. In a clinical study, patients were analyzed for PASI-score, serum levels vitamin D and regulatory T-cells percentage. Using non parametric Spearman coefficient test to assess correlation between serum levels of vitamin D and the single variables of disease, this study found a positive association between vitamin D and Tregs population (*p* < 0.001), and an inverse correlation between vitamin D and PASI-score (*p* = 0.04) [77]. The effects of maxacalcitol, a vitamin D3 analogue, and betamethasone valerate (BV) steroid lotion, confirmed the effects of vitamin D on the differentiation of T-cells with suppressive phenotypes in an imiquimod (IMQ)-induced psoriasiform skin inflammation animal model. The authors report that maxacalcitol and BV reduced the MHC Class II^+^ inflammatory cell infiltrate and down-regulated IL-17A, IL-17F, IL-22, IL-12p40, TNF-α and IL-6 mRNA expression levels in the inflamed mouse ski. Maxacalcitol alone downregulated IL-23p19 expression, and increased Foxp3^+^ T-cell infiltrations and IL-10 expression. Of note, adoptive transfer of Tregs from maxacalcitol-treated donor mice improved IMQ-induced inflammation more than of Tregs from a BV-treated donor group [78]. Many data in the literature, from humans and mouse models, report that vitamin D induces myeloid dendritic cells with a tolerogenic phenotype responsible for the differentiation of CD4 CD25^+^ Tregs from naive T cells [79,80,81].

#### 2.4.4. Topical Therapies

Topical therapies based on glucocorticoids (GC) and calcipotriol are usually sufficient to manage mild and moderate psoriasis [82]. GCs produce anti-inflammatory effects through GC receptors (GR) and by acting on specific target genes, inhibiting several cytokines [83]. Calcipotriol exerts its effect by binding to the nuclear vitamin D3 receptor [84]. The anti-inflammatory effects of calcipotriol are inferior compared with those of GCs, but an incremented effect is seen with a combinatory therapy [85]. Keijsers and co-workers showed that topical calcipotriol/betamethasone treatment for eight weeks decreased the number of Tregs in psoriatic lesions and the expression of Foxp3 in the skin and PBMCs [86]. Minna E. Kubin et al. confirmed that a combination therapy down-regulated the expression of TNF-α, IL-23, IL-17A, S100A7, CCL20 and interferon-γ in the skin and TNF-α, IL-6, IL-23A, T-bet and IFN-γ in PBMCs. Calcipotriol/betamethasone, but not betamethasone alone, down-regulated expression of Foxp3 in both skin and PBMCs [87].

#### 2.4.5. Oral Small Molecules

##### Dimethyl Fumarate (DMF)

The European Medicines Agency [88] approves this drug for the treatment of psoriasis patients as of 2017, as an oral formulation. Studies in vitro show that DMF promotes oxidative stress reducing vitality of conventional T-cells but not Tregs. An increased expression on Tregs of cell surface-reduced thiols or thioredoxin-1 [89], protect Tregs from oxidative stress, mediated by DMF. The anti-psoriatic effect of DMF favors Tregs survival but not Th17 expansion [90]. In psoriasis patients, DMF treatment increased Tregs frequency and decreased Th17 cells, confirming in vitro data [91].

##### Sotrastaurin

A clinical study in psoriasis patients, using the pan-protein kinase C (PKC) inhibitor sotrastaurin (AEB071), showed a reduction of psoriasis clinical severity [92]. Currently, sotrastaurin is in phase II clinical trial studies for psoriasis [93]. Sotrastaurin blocks more than one PKC isoform. The latter belongs to a sub family of PKC calcium-independent and is most abundant in T-cells [94]. The activation of T-cells by CD28 and TCR promotes PKC-theta activation and translocation into the membrane at site of immunological synapse (IS), leading to the activation of NF-kB. Inhibition of PKC-theta restored activity of defective Tregs from RA patients and enhanced protection of mice from inflammatory colitis [95]. Moreover, studies in PKC-knockout mice have shown that PKC-theta is required for productive Th2 [96] and Th17 [97] responses but not for Th1 responses. In particular, Xuehui He et al. confirmed that sotrastaurin prevented TCR/CD28-induced T-cell activation and pro-inflammatory cytokine production, and enhanced Tregs response [98].

##### Janus Kinase (JAK) Inhibitors

This class of inhibitors is apparently restoring Tregs activity in psoriasis. The binding of cytokines to their receptors enables the activation of the JAK/STAT signaling pathways. This happens for IL-6, IFN-γ, IL-22, and IL-21, all involved in psoriasis. JAK inhibitors may thus suppress the effects of inflammatory cytokines involved in the disease [99]. The same (JAK) inhibitors are indicated for treatment of PsA, as the Food and Drug Administration approved the inhibitor tofacitinib, whereas the JAK1 inhibitor upadacitinib is approved in Japan. In a model of hepatitis in mice, induced by concanavalinA (ConA), tofacitinib increased the ratio of Tregs/Th17 cells as detected not only in the mouse liver but also in the spleen, which is representative of the situation in the peripheral regions [100]. An in vitro study reported that tofacitinib suppresses T-effector functions but preserves activity of CD4 CD25^bright^ Tregs. This may explain its capacity to increase the Tregs/Th17 ratio [101].

##### Methrotrexate (MTX)

MTX, a folic acid analogue, is another treatment for psoriasis that is able to inhibit the activation of lymphocytes and macrophages, thus modulating cytokines, and inhibiting neutrophil chemotaxis [102]. MTX monotherapy determined, after 15 weeks of treatment, an increase in the percentages of Th2/Treg cells and a concomitant decrease of Th1 and Th17 cells [103]. In a study by K. Yan et al., Tregs and effector T-cells were isolated from blood of patients with psoriasis and healthy controls. In psoriasis patients, Tregs had a decreased immune suppressive function and a reduced expression of CD73, as compared to the healthy controls. Both IL-17 and IFN-γ were significantly upregulated in psoriasis, implying that T effector cells in the tissues possessed an aberrant secretion capacity of Th1/Th17 cytokines. The authors observed that, in patients, MTX treatment induced a significant growth inhibition of T effector cells. The production of IL-17 and IFN-γ by Tregs was also reduced, suggesting that MTX restores the function of Tregs and restrains the proliferation of T effectors in psoriasis patients. The authors analyzed CD73 expression by flow cytometry, and the phosphorylation of AMPK and mTOR by western blot. In all patients, MTX treatment reversed down-regulation of CD73, activated AMPK and inactivated mTOR [41]. In contrast to conventional resting T-cells, Tregs were found to express both CD39 and CD73 at high levels. These surface nucleosidases enzymatically active possess immunosuppressive properties on effector T-cells by negative feedback responses via the adenosine receptor (A2AR) but also via low-affinity receptors (like the A2B-adenosine receptor, A2BR). A2AR is ubiquitously expressed in a wide variety of immune cells including T-cells, B cells, NK cells, NKT cells, macrophages, dendritic cells, and granulocytes; A2BR plays a distinctive role in controlling inflammation, for example, via the induction of a tolerogenic antigen presenting cells (APC), via an alternative activation. Upon interaction with A2AR, adenosine is responsible for the inhibition of T-cell activation. Moreover, immunosuppressive activity may be further enhanced by adenosine, which induces Tregs, promoting tolerogenic antigen-presenting cells (APC) and myeloid-derived suppressor cells (MDSC) activities [104].

Other potential and promising targets, which can be useful in the regulation of Tregs, or to reset the imbalance of T-helper/Tregs in psoriasis, are under investigation; among them, IL-2 at low dose, histone deacetylase inhibitors sodium butyrate, STAT3 inhibitors, probiotics and T-cell based therapies [105,106,107,108,109,110].

## 3. Tregs Specific for the Autoantigen LL37 Can Be Present in Humans

### 3.1. Introduction and Rational

The data and experiments illustrated above, about the capacity of diverse treatments options to restore the number of Tregs and/or their effector functions in psoriasis, indicate that Tregs are important to counteract inflammation in this chronic skin disease. Studies on the specificity of Tregs in psoriasis are lacking, although it has long been known that the disease is T-cell driven. The specificity of psoriasis T-cells driving inflammation in the skin has been elusive for a long time. Previous studies demonstrated that specificity was directed towards keratins [111]. Later, we discovered that the antimicrobial peptide (AMP) cathelicidin LL37 is an autoantigen in psoriasis [50], and another study found that ADAMTSL5 is also an autoantigen in psoriasis [112]. In the past, we have identified LL37-specific T-cells by using peptide-MHC-tetramers. This approach, together with epitope mapping and cloning of the T-cells, identified the most immunogenic parts of LL37 and the restriction molecules for presentation to T-cells (there were several HLA-class I and class II alleles involved in the recognition by CD4 and CD8 T-cells, among which were HLA-DR7, HLA-DR11, HLA-DR4, and HLA-Cw6) [50,113]. Usually, healthy donors (HD) do not respond to LL37 or LL37-derived shorter antigenic peptides in T-cell proliferation assays [50,113]. Occasionally, low and rare proliferation can be detected in HD, which is not significant as compared to the psoriasis group. With this in mind we tried to check, in HD with a low LL37 T-cell proliferative response, whether LL37-specific T-cells existed and whether these cells belong to the Tregs compartment (the strategy and methods we used for this analysis is reported in Section 4 (see below).

### 3.2. Results

#### 3.2.1. LL37-Specific T-Cell Clones Can Be Obtained also from HD

Knowing the HLA-typing of some HD, we managed to obtain a T-cell clone specific for LL37 in a HLA-DR11-positive individual (see Methods). This clone was stained by a peptide-MHC-tetramer of the MHC haplotype HLA-DR11 linked to peptide P6 (that we call DR11-P6 tetramer), but not by a second peptide-MHC-tetramer of HLA-DR11, and loaded with a different LL37-derived peptide (P4), referred to as DR11-P4 tetramer.

#### 3.2.2. Expanded Tregs Are Stained by LL37 Specific Peptide-MHC-Tetramers

Next, we isolated Tregs from the PBMCs of the same donor by using magnetic isolation kit and obtained Tregs that were expanded for three weeks in the presence of a high dose of human recombinant (hr) IL-2 (according to previous published protocols, see methods) [114]. Once Tregs were expanded by repeated stimulations (phenotype of the obtained Tregs is reported in Appendix A), in the presence of high dose hrIL-2 (300 U/mL), we obtained T-cells that could be, in part, stained by the same peptide-MHC-tetramer DR11-P6, but not by the control peptide-MHC-tetramer DR11-P4 (Figure 1). Figure 1 shows staining of clone T (Figure 1, Clone T) and of Tregs (Figure 1, Tregs), derived from the same donor.

#### 3.2.3. Expanded Tregs Suppress Responder T-Cells with the Same Specificity In Vitro

At this stage, we performed a suppression assay to see whether the recovered Tregs were able to suppress activation of the clone T and its production of IFN-γ after over-night culture (Figure 2a). The gating strategy for this assay is shown in Appendix A. To distinguish responder T-cells of the clone (T) from the Tregs, we pretreated the latter with CSFE to exclude these cells from the analysis of IFN-γ production in response to the LL37 antigenic peptide (P6). The data in Figure 2a (and Appendix A) show that clone T responded to LL37-P6 peptide by producing IFN-γ, but the presence of Tregs reduced this production. The Tregs were not able to respond producing IFN-γ in response to P6, as shown when they were cultured with the APC alone presenting the peptide P6, in the absence of the T-cells derived from the clone (T). We also performed similar experiments during a short-term culture (2 h), where we found inhibition of the capacity of clone T to produce IL-2 in the presence of APC and P6 when Tregs were present into the cultures (Figure 2b). IL-2 production from the clone was assessed by catch assay (see methods).

#### 3.2.4. Discussion

This overview of the role of Tregs in psoriasis and our own experimental findings suggest that autoantigen specific, in this case LL37 specific T-cells, can be part of the physiological Tregs pool in HD. A person with T-cells responding to LL37 may have Tregs that are also specific for LL37 (as shown here by using peptide-MHC-tetramers specific for LL37-epitopes). Such T-cells, which in our hands were able to act as regulatory cells in vitro by suppressing cytokine production by their own reactive LL37-specific T-cells, could be natural Tregs or induced Tregs. The evidence that LL37 is expressed not only in various organs but also in the thymus [115] could support the hypothesis that natural Tregs, specific for LL37, are present physiologically in HD. Expression of LL37 in the thymus may allow the deletion of LL37-specific T-cells by negative selection, or the selection of T-cells, which such specificity, endowed with regulatory/suppressive activity, which regulate the immune responses [116].

#### 3.2.5. Limitations

We cannot exclude that the cells of the T-cell clone T were obtained after an in vitro priming of the LL37-specific T-cells during the cloning procedure. However, these results are reported here to discuss the possible presence of T-cells specific for LL37 in vivo in HD. The limitation of this approach is that we ignore whether this is true for every HD. The experimental data presented here should, therefore, be viewed as an in vitro model of T-cell suppression, and not as a formal demonstration that both responder and regulatory T-cells specific for the autoantigen LL37 exist in vivo in HD.

#### 3.2.6. Conclusions

Tregs certainly play an important role in psoriasis, a disease in which they are dysfunctional. Many psoriasis treatments seem to exert an effect on Tregs, which in some cases acquire again their lost regulatory functions, as shown by our review of the literature. The presence of Tregs specific for LL37 and other T-cell autoantigens in psoriasis can be addressed in the same way reported here, by using peptide-MHC-tetramers. The mechanism of immune-suppression could be also addressed. The experiments are not easy to conduct, as one should perform HLA-typing of different HD and use several peptide-MHC-tetramers to stain the Tregs and identify the correct cells. One issue to address, using improved protocols like those presented here, is whether antigen-specific Tregs are altered in psoriasis and whether effectors autoreactive T-cells are derived from existing Tregs or are newly formed. Similar experiments can be useful to address whether the therapies used for psoriasis, mentioned here, can induce recovering of dysfunctional T-cells or imply a de novo induction of Tregs, or elimination of responder autoreactive T-cells.

## 4. Material and Methods

This review represents a narrative review. A search was conducted in the scientific literature on PubMed and Google Scholar, searching by using the following keywords: “Tregs in psoriasis” “Tregs and psoriasis treatments or psoriasis medications”, considering also the “related articles” in PubMed. We placed no time limit on the research performed and we included epidemiological studies, animal models, and in vitro culture models.

### 4.1. Detailed Methods for the Experimental Part of this Study

#### 4.1.1. Purification of Human Tregs and Their Expansion

Human Tregs were enriched from the peripheral blood of a healthy volunteer, whose cells show low proliferation to LL37 and to peptide P6 (see below) of LL37. Cells were obtained with full informed consent and ethical approval. PBMC were separated by Ficoll-Hypaque (Pharmacia Fine Chemicals, Uppsala, Sweden) density gradient centrifugation and total CD4 T cells were isolated using the CD4^+^CD25^+^ CD127^dim/−^ Regulatory T Cell Isolation Kit II, human (Miltenyi Biotech, San Diego, CA, USA). Expansion of the cells was performed over three weeks in the presence of a high dose of hrIL-2 (300 U/mL) in complete medium containing RPMI 1640, 10% heat-inactivated human serum, HS, (Gibco -Thermo Fisher Scientific, Waltham, MA, USA), 2 mM L-glutamine, 10 U/mL penicillin and 100 μg/mL streptomycin (Sigma-Adrich, St. Louis, MO, USA), according to a modified protocol previouly published [115]. Tregs were screened for expression of markers CD38, CD25, CD127 by flow cytometry using a Gallios flow cytometer (Beckman Coulter, CA, USA), after isolation and after expansion.

#### 4.1.2. Generation of T-Cell Clones

For the generation of the T-cell clone T, specific for LL37 P6, PBMC were stimulated (2 × 10^6^ cells/mL) P6 of LL37, for 7 days and re-stimulated after 10 days with LL37-peptide P6 (10 μg/mL, seq: VQRIKDFLRNLVPRT), (synthesized by Biomatik, Kitchener, ON, Canada), and autologous-γ-irradiated-(30Gy)-PBMC in complete medium in the presence of human/recombinant(hr)IL-2 (Boehringer-Mannheim, Indianapolis, IN, USA). T-cell line-specificity was analyzed by using peptide-pulsed (10 μg/mL) autologous-irradiated PBMC (30Greys) or lymphoblastoid-cell lines (B-LCL) (150Greys) and peptide P6 or P4 (P4: IGKEFKRIVQRIKDF also derived from LL37) and control REV LL37 peptide, as previously described [50]. BrdU was added at day 4, T cells were analyzed by flow cytometry as described [50]. Cells were cloned by limiting dilution in Terasaki plates (Nunc Microwell, Sigma-Aldrich, St. Louis, MO, USA) in the presence of allogenic-irradiated (10^4^) PBMCs activated by phytohaemagglutinin (1 μg/mL, PHA, Murex, Cedex, France) at 0.5 cells per well, as previously described [50]. 100 U/mL of hrIL-2 were added. Expanded clones reactivity and HLA restriction were analyzed using HLA-DR-matched-homozygous B-LCLs (ATCC, Virginia, USA), pulsed with peptide antigen or control peptide antigens, as previously described [50].

#### 4.1.3. T-Cell Suppression Assay

T-cell clone T (5 × 10^4^ cells per well, in duplicates) was cultured with autologous-irradiated PBMCs (10^5^ cells per well), and without or with peptide P6 at 10 μg/mL, with or without Tregs (ratio 1.1), overnight. The day after, we performed intra-cellular cytokine staining for IFN-γ on gated responder T-cell clone (T) and on Tregs (identified as CSFE-positive T-cells). Fluorescence was analyzed by a Gallios flow-citometer (Beckman Coulter, CA, USA), and FCS files were analyzed with FlowJo 7.5 software (TreeStar Inc., Ashland, OR, USA). For IL-2 detection in suppression assay, the T-cell clone was cultured with autologous-irradiated PBMCs with or without antigen P6 for two hours in the presence/absence of Tregs, as above and stained with the IL-2 Secretion Assay—Detection Kit (PE), human, (Miltenyi Biotech, Gaithersburg, MD, USA) for IL-2 and for anti-surface molecules (anti-CD3, anti-CD4) to detect IL-2 secretion in the presence or absence of Tregs (1:1 ratio). The latter were colored with CSFE (Sigma). For CSFE labeling of T regs (2 × 10^5^ cells) were treated with 5 mM CSFE in the dark for 10 min in a small volume (100 mL) and, at the end of incubation, cold complete medium was added (four volumes), putting the cells in ice for 5 min. Cells were centrifuged and washed twice in complete medium (medium with 15% of HS).

#### 4.1.4. Peptide-MHC-Tetramer Staining

The following peptide-MHC-tetramers were used: P6-DR11-tetramer (called P6 tetramer) (P6: VQRIKDFLRNLVPRT) and P4-DR11 tetramer (P4: IGKEFKRIVQRIKDF), called control tetramer, both synthesized by TC Metrix, Epalinges, CH. Staining was performed at 37 °C for 40 min, followed by staining for CD4 (4 °C, 20 min). For the detection of tetramer-positive cells in Tregs and clone T, before flow cytometry acquisition, cells were labeled with 7-AAD to exclude dead cells and avoid unspecific staining.

#### 4.1.5. Antibodies (Mabs)

Mabs to CD4, CD3 conjugated with various fluorochromes (FITC, phycoerythrin (PE), peridinin-chlorophyll-protein (PerCp Cy5.5), or allophycocyanin (APC), were from BD Biosciences or eBiosciences (San Diego, CA). PE- or APC-CD38, PE-CD127, FITC-CD25 mabs were purchased from BD Biosciences, eBiosciences, Novus Biologicals (Littleton, CO, USA), R&D (Minneapolis, MN, USA). Appropriate isotype-matched controls were purchased from the same companies. PerCp-7-AAD was from BD Pharmingen.

Foxp3 staining on cultured Tregs was done by intracellular staining, using an APC-anti-FoxP3 (PCH101). Staining was performed with an eBioscience Fix/Perm kit under the manufacturer’s directions.

#### 4.1.6. Statistical Analysis

Data shown are means ± SEM, where indicated. Statistical comparison in Tregs suppression assays was performed using a two-tailed paired-samples Student’s *t* tests. Statistical significance was set at *p* < 0.05.

## Figures and Tables

**Figure 1 ijms-24-04348-f001:**
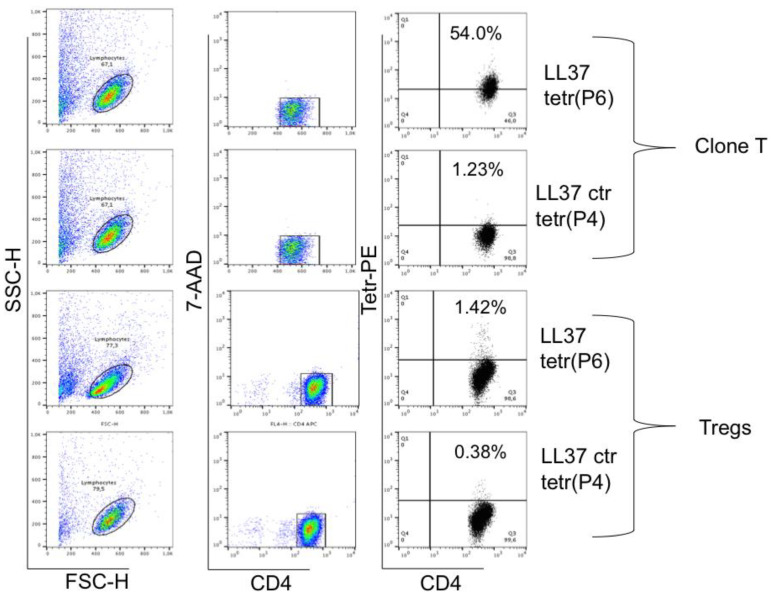
Enriched Tregs isolated from PBMCs are LL37-specific. Clone T, specific for LL37 P6 was stained with peptide-MHC-tetramer DR11-P6 (see methods), upper panels, and peptide-MHC-tetramer P4 (control tetramer, ctr). Tregs were isolated from PBMCs from the same HD and cultured for 3 weeks in the presence of 300 U/mL of hrIL-2. These cells were stained with the same peptide-MHC-tetramer and control peptide-MHC-tetramer as for the clone, lower panel. Experiments were repeated three times, with the same donor.

**Figure 2 ijms-24-04348-f002:**
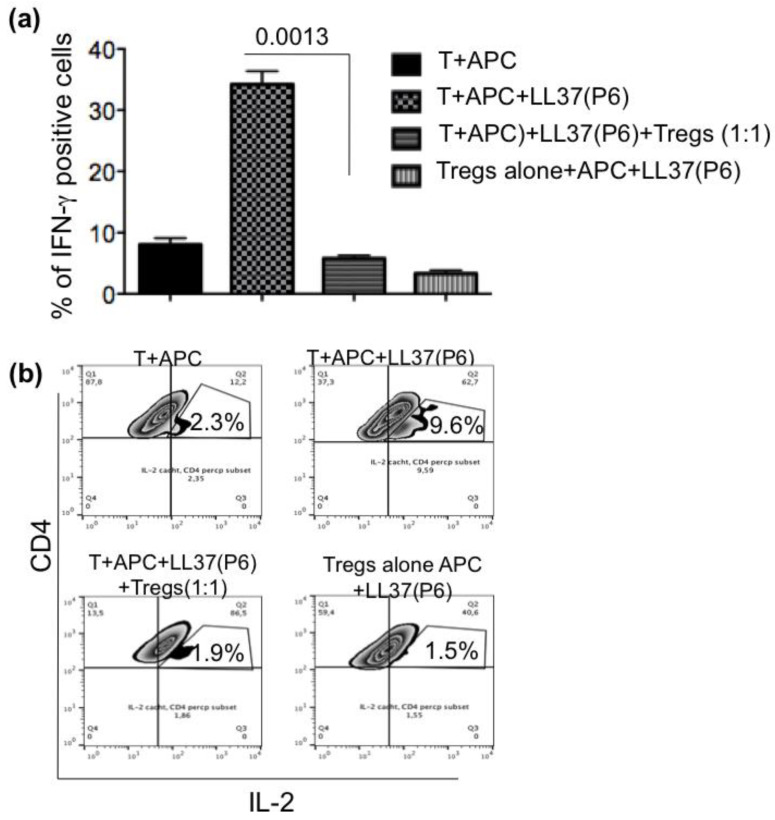
Tregs LL37-specific can be present in HD and be suppressive in vitro. (**a**) T-cell clone (T), derived from a HD and specific for LL37 (peptide P6), was cultured either alone, or with APC alone or APC presenting its cognate antigen LL37 P6, or in the presence of Tregs isolated and enriched from PBMC of the same HD (and pretreated with CSFE) (as in Appendix A), also in the presence of APC and antigen (P6), overnight. Tregs were also cultured, alone, with APC and antigen P6. APC were irradiated to block their proliferation. The day after cells were harvested and IFN-γ production was assed, by intracellular staining and flow cytometry in gated T-cells derived from clone T (which were CSFE negative and distinct from the CSFE^+^ Tregs) as described in Methods. Experiments were repeated two times (in duplicates), with the same donor. (**b**) The same type of experiment was performed with clone T in the culture conditions of (**a**), but for 2 hours, detecting IL-2 production by the clone T by catch assay (see methods).

## Data Availability

Data are available upon reasonable request.

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
