# Peer review of "Autoreactive T-Cells in Psoriasis: Are They Spoiled Tregs and Can Therapies Restore Their Functions?"

_ijms, 2023, doi:10.3390/ijms24054348_

Round 1

Reviewer 1 Report

Congratulations to the authors for a very thorough analysis of the existing literature with regards to the role of Treg in psoriasis inflammations. However, I have some comments:

1.     It has been shown that Tregs are part of the TRM (Tissue Resident Memory cells) compartment. (Samat AAK, van der Geest J, Vastert SJ, van Loosdregt J, van Wijk F. Tissue–Resident Memory T Cells in Chronic Inflammation—Local Cells with Systemic Effects? Cells. 2021; 10(2):409. )

2.     It is worth to mention about the role of PELI1-mediated NF-kB signaling is involved in Treg dysfunction found in psoriasis. (Sung Hee Kim, et al. Pellino-1 promotes intrinsic activation of skin-resident IL-17A-producing T cells in psoriasis,Journal of Allergy and Clinical Immunology, 2023,)

3.     There is no separate paragraph "Discussion", short summary "Conclusions" and "Limitations".

Author Response

Dear reviewer,

We thank you for your careful evalutation.

Our reply is attached.

Best regards

Loredana Frasca

Reviewer 2 Report

The submited manuscript integrates a review of the topic and some experimental work. They should be clearly separated, and the Material and Methods section should include some detail on the search strategy of the (scoping?) narrative review.

Overall, sections 2.1 and 2.2 should be revised to improve logical order and avoid repetition.

Further explanation should be provided for the CD73 pathway commented on lines 118-121.

The order of presentation of drugs in section 2.3 seems rather arbitrary. Please organize according to standard usage and amount of information: biologics, small molecules, phototherapy, and topicals, or the other way round.

Section 2.9 (should be 3) apparently includes original work, and this should be clearly stated from the begining. This long tirade should be divided in sections and paragraphs as an original manuscript, with introduction, material and methods, results and discussion. A final Conclusion section is required.

There are some minor mistakes in numbering of paragraphs (e.g. those following 2.3  on line 143, or section 2.9), number concordance of verbs and syntax that should be corrected (e.g. more instead of several on line 57, has instead of have on line 63, count-part instead of counterpart on line 74, situation instead of situations on line 84, etc.

Author Response

Dear Reviewer,

We thank you for the careful evalutation of our manuscript.

We have attached our reply below.

Best regards

Loredana Frasca

Round 2

Reviewer 1 Report

I have no comments now

Reviewer 2 Report

Overall, my comments have been addressed. Please revise the manuscript for some English spelling/syntactic errors.